# Overexpression of *PtrMYB121* Positively Regulates the Formation of Secondary Cell Wall in *Arabidopsis thaliana*

**DOI:** 10.3390/ijms21207734

**Published:** 2020-10-19

**Authors:** Ying Liu, Jiayin Man, Yinghao Wang, Chao Yuan, Yuyu Shi, Bobin Liu, Xia Hu, Songqing Wu, Taoxiang Zhang, Chunlan Lian

**Affiliations:** 1International Joint Laboratory of Forest Symbiology, College of Forestry, Fujian Agriculture and Forestry University, Fuzhou 350002, China; yingliu1112@163.com (Y.L.); jiayinmanjiayin@163.com (J.M.); 18438616077@163.com (Y.W.); Yuanchao507@163.com (C.Y.); yuevan2014@163.com (Y.S.); lake-autumn@163.com (X.H.); 2Fujian Colleges and Universities Engineering Research Institute of Conservation and Utilization of Natural Bioresources, College of Forestry, Fujian Agriculture and Forestry University, Fuzhou 350002, China; liubobin@fafu.edu.cn; 3Key Laboratory of Integrated Pest Management in Ecological Forests, Fujian Province University, Fujian Agriculture and Forestry University, Fuzhou 350002, China; dabinyang@126.com; 4Asian Natural Environmental Science Center, The University of Tokyo, 1-1-8 Midori-cho, Nishitokyo, Tokyo 188-0002, Japan

**Keywords:** *PtrMYB121*, secondary cell wall, transcription factor, lignin, cellulose

## Abstract

MYB transcription factors have a wide range of functions in plant growth, hormone signaling, salt, and drought tolerances. In this study, two homologous transcription factors, *PtrMYB55* and *PtrMYB121,* were isolated and their functions were elucidated. Tissue expression analysis revealed that *PtrMYB55* and *PtrMYB121* had a similar expression pattern, which had the highest expression in stems. Their expression continuously increased with the growth of poplar, and the expression of *PtrMYB121* was significantly upregulated in the process. The full length of *PtrMYB121* was 1395 bp, and encoded protein contained 464 amino acids including conserved R2 and R3 MYB domains. We overexpressed *PtrMYB121* in *Arabidopsis thaliana*, and the transgenic lines had the wider xylem as compared with wild-type Arabidopsis. The contents of cellulose and lignin were obviously higher than those in wild-type materials, but there was no significant change in hemicellulose. Quantitative real-time PCR demonstrated that the key enzyme genes regulating the synthesis of lignin and cellulose were significantly upregulated in the transgenic lines. Furthermore, the effector-reporter experiment confirmed that *PtrMYB121* bound directly to the promoters of genes relating to the synthesis of lignin and cellulose. These results suggest that *PtrMYB121* may positively regulate the formation of secondary cell wall by promoting the synthesis of lignin and cellulose.

## 1. Introduction

The most abundant biological resource produced by woody plants is wood, which is widely used in traditional industries including lignum, papermaking, and pulping. Wood is also one of the important materials used in the production of bioethanol [1]. Formation of wood involves formation of mother cells of secondary xylem differentiated from the vascular cambium, elongation of cells, deposition of secondary cell wall (SCW), and death of programmed cells [2]. Wood mainly contains SCW, which is synthesized in special types of cells, such as vessel elements and fibroblasts, and mainly consists of cellulose, lignin, and hemicellulose [3]. SCW participates in many important biological processes, such as water and nutrient transport, mechanical support of plant tissues and organs, and stress responses. Therefore, research on the molecular mechanisms of SCW formation is a hot topic.

Up to now, biosynthesis of the main SCW components has been clearly studied. Cellulose is synthesized by cellulose synthase (CesA) complexes, which can be divided into the following two types: type I which is mainly involved in the formation of primary walls and type II which is involved in fiber, vessels, and xylem parenchyma synthesis of SCW. Bioinformatics analysis has revealed that there were 10 CesA genes in Arabidopsis [4], 18 CesA genes in poplar [5], and several CesA genes in rice [6]. Lignin biosynthesis is more complicated than cellulose synthesis and is mainly regulated by phenylalanine ammonia-lyase (PAL), cinnamate 4-hydroxylase (C4H), 4-coumarate-CoA ligase (4CL), p-coumarate 3-hydroxylase (C3H), caffeoyl shikimate esterase (CSE), shikimate/quinate hydroxycinnamoyl transferase (HCT), caffeoyl CoA3-O-methyltransferase (CCoAOMT), cinnamoyl CoA reductase (CCR), ferulate 5-hydroxylase (F5H), caffeic acid O-methyltransferase (COMT), and cinnamyl alcohol dehydrogenase (CAD) [7,8]. Hemicellulose is a type of heteropolysaccharide, and the skeleton of most hemicellulose is single. Most hemicellulose main chains are synthesized by cellulose synthase-like (CSL) genes, including the CSLC, GT34, GT37, and GT47 families [9,10].

As compared with biosynthesis, transcriptional regulation of SCW has not been completely understood. On the basis of increasing data regarding whole genome sequences from different plants, a great number of transcription factors (TFs) that regulate SCW formation have been identified, such as the NAC, MYB, and WRKY families. Previous reports have indicated that there was a transcriptional regulatory network among these families for regulating biosynthesis of SCW [11]. Within this network, the NAC TFs are the first-level master switch genes that regulate ectopic deposition of SCW. These switch genes, named as VNDs (vascular-related NAC domains), were first identified in differentiation of xylem and fiber of Arabidopsis [12]. Overexpression of VNDs in Arabidopsis resulted in ectopic deposition of vessel elements of the cell wall and inhibited the formation of protoxylem and epigenetic xylem [13,14]. To date, some NAC TFs have been found that function as master switches in various species of angiosperms, such as poplar, alfalfa, rice, eucalyptus, and birch, which suggests that these first-level master switch genes could conservatively regulate the SCW synthesis in vascular plants [15,16,17,18,19,20].

Another important gene family that regulates SCW biosynthesis is R2R3-MYB TFs, which belong to the second-level switch genes. Of these members, MYB46 and MYB83 in Arabidopsis are the key genes [21,22], and previous studies have shown that they could promote the thickening of SCW in vessels and fibers [21,22,23]. Transcriptional activation and electrophoretic mobility shift assays (EMSAs) have shown that a variety of downstream TFs and structural genes of SCW biosynthesis could be directly activated by *AtMYB46* and *AtMYB83* [22]. Similar to the first-level master switch genes, the functions of these MYB TFs in SCW regulation were also conserved, and have been identified in poplar, eucalyptus, pine, rice, maize, and switchgrass [24,25]. Meanwhile, *AtMYB61* was founded to be a multifunctional TF in Arabidopsis. It specifically regulates lignin biosynthesis [26,27], affects the trichome formation and root development [28], regulates the stomata development [29], and also has effects on the coloring of seed coat [30]. Furthermore, *AtMYB61* also links the transcription regulation of multiple aspects of resource allocation [26].

Poplar is one of the most widely cultivated fast-growing trees around the world and important to the wood industry. As a woody model plant, the whole genome of several poplar species [31,32,33] has been sequenced and the genetic transformation system has also been established. MYB TFs in poplar have been studied less than other model plants, and the regulatory mechanisms of SCW formation is still unclear. There are at least 194 R2R3-MYB transcription factors in the *Populus tricocharpa* (*P. tricocharpa*), but less than 10% have been isolated and characterized for function analysis [34]. Therefore, it is essential to identify and analyze the functions of MYB TFs in SCW formation in order to increase wood yield and improve quality, with the development of transgenic technologies in poplar.

In this study, we isolated a MYB TF *PtrMYB121* from *P. tricocharpa*, and its function was investigated by tissue expression analysis, overexpression in *Arabidopsis thaliana* (*A. thaliana*), qRT-PCR, and transcriptional activation assays. The results indicate that *PtrMYB121* can activate the expression of structural genes involved in the biosynthesis of lignin and cellulose and promote the accumulation of lignin and cellulose.

## 2. Results

### 2.1. Phylogenetic Analysis and Amino Acid Sequence Alignment of PtrMYB121

In order to characterize some poplar MYB TFs in the process of SCW formation, the amino acid sequence of *AtMYB61* was used to blast and multiple alignment. Two pairs of R2R3-MYB homologous genes (*PtrMYB55/121* and *PtoMYB170/216*) of *AtMYB61* were isolated (Figure 1A), and *PtoMYB170/216* were demonstrated to specifically regulate lignin biosynthesis [35,36]. The sequence alignment results showed that the R2 and R3 domains of *PtrMYB55/121* were highly conserved with *AtMYB61* and *PtoMYB170/216*, and only had differences with a few bases (Figure 1A). The amino acid residues in the R2 MYB domain were changed from histidine to tyrosine, and from asparagine to arginine and serine, while the amino acid residues in the R3 MYB domain were changed from alanine to threonine, and from isoleucine to leucine (Figure 1A). Meanwhile, we also found that the other regions of *PtrMYB55/121* varied greatly, especially the C-terminus region (Figure 1A). Hence, we constructed a phylogenetic tree with *PtrMYB55/121* and some MYB TFs that have been reported to play key roles in SCW biosynthesis from different species. As shown in Figure 1B, these MYB TFs were divided into three distinct clusters, and *PtrMYB55/121* were distributed on an independent branch with *PtoMYB170/216* and *AtMYB55/61* (Figure 1B). The results suggest that *PtrMYB55/121* should participate in the SCW formation but may have functional differentiation with other branches.

### 2.2. Tissue Expression Pattern Analysis of PtrMYB55 and PtrMYB121

To analyze the expression patterns of *PtrMYB55* and *PtrMYB121* in different tissues, total RNA from various tissues and organs (young leaves, old leaves, young stems, old stems, and roots) of three-month-old *P. trichocarpa* seedlings was extracted for qRT-PCR analysis. The qRT-PCR analysis showed that *PtrMYB55* and *PtrMYB121* had similar expression patterns, and transcript abundance of *PtrMYB55* and *PtrMYB121* could be detected in all organs and tissues (Figure 2). They had the highest expression level in old stems, followed by young stems, which was consistent with SCW-related studies (Figure 2). However, the expression of *PtrMYB55* and *PtrMYB121* was lower in young leaves, old leaves, and roots than in stems (Figure 2). Combined with the results in Figure 1, we speculate that *PtrMYB55* and *PtrMYB121* had functional redundancy in SCW formation. *PtrMYB121* was used for further research due to its preferential expression in stem.

### 2.3. Overexpression of PtrMYB121 in A. thaliana

To analyze the function of *PtrMYB121* in SCW biosynthesis, we overexpressed its full-length coding sequence (CDS) in wild-type *A. thaliana*. Then, we obtained 11 hygromycin-resistant *A. thaliana* lines and analyzed the function of *PtrMYB121* in the T2 homozygous generation. A part (783bp) of the CDS sequence of *PtrMYB121* was amplified by PCR using gene-specific primers (Appendix A), and nine lines were confirmed to be the transgenic Arabidopsis seedlings (Figure 3A). qRT-PCR analysis was used to select candidate lines of *PtrMYB121*-overexpressing transgenic Arabidopsis for later functional verification, and the results showed that L1 (14.6-fold), L4 (11.3-fold), and L6 (9.6-fold) had relative higher expression levels of *PtrMYB121* as compared with other lines (Figure 3B).

### 2.4. Overexpression of PtrMYB121 Promotes the Accumulation of Lignin and Cellulose

There was no significant difference in the phenotypes between the wild-type (WT) and *PtrMYB121*-overexpressing transgenic Arabidopsis during the 40-day growth process. We also measured the trichome density, plant height, plant diameter, and leaf area per rosette leaf of the 40-day-old WT and *PtrMYB121*-overexpressing Arabidopsis, and they had no significant difference (Appendix A). The three independent lines (L1, L4, and L6) had similar phenotypes, and L1, which had the highest expression, was used for histochemical staining. Anatomical cross-sections of inflorescence stems were collected and made from the bases of 40-day-old wild-type and transgenic Arabidopsis plants, and used for toluidine blue O (TBO) staining. The results showed that overexpression of *PtrMYB121* caused the significant increase in cell layers (7–16 layers, Figure 4E,F and Table 1) of xylem as compared with WT stems (4–9 layers, Figure 4A,B and Table 1).

To quantitatively analyze the changes in xylem of *PtrMYB121*-overexpressing transgenic Arabidopsis, we further made the anatomical cross-sections of inflorescence stems and took pictures with an optical microscope system. Then, Image J software was used for calculating the cell layer number and radial width of xylem. Both WT (WT-1) and *PtrMYB121*-overexpressing Arabidopsis (L1) contained three independent samples, and every sample had 30 slices for analysis. The cell layers of xylem increased 89.73–120.84% in *PtrMYB121*-overexpressing Arabidopsis (Figure 4E,F and Table 1). The xylem area of *PtrMYB121*-overexpressing transgenic Arabidopsis (Figure 4E,F and Table 1) was 38.35–64.12% wider than that of inflorescence WT stems (Figure 4B and Table 1). Statistical analysis with one-way ANOVA showed the cell layer and radial width of xylem in inflorescence stems had significant differences between WT and *PtrMYB121*-overexpressing transgenic plants (Table 1), but the three independent samples among WT or *PtrMYB121*-overexpressing transgenic Arabidopsis had no significant difference (Table 1). We also used a fluorescence microscope to observe the autofluorescence of lignin in the bases of inflorescence stems of WT and *PtrMYB121*-overexpressing transgenic lines. Overexpression of *PtrMYB121* (Figure 4G,H) caused the wider xylem and ectopic deposition of lignin as compared with the WT (Figure 4C,D).

To quantify the changes of SCW components, we measured the contents of lignin, cellulose, and hemicellulose. The acetyl bromide (AcBr) method was used to estimate the total lignin content [37], while the Van Soest method was used to measure the cellulose and hemicellulose content [38]. The results revealed that lignin content in the stems of *PtrMYB121*-overexpressing transgenic lines was significantly increased 14.37–51.49% as compared with the WT plants (Figure 5), while the cellulose in transgenic Arabidopsis was increased 20.42–31.33% (Figure 5). However, there was no difference between the content of hemicellulose in WT and *PtrMYB121*-overexpressing lines (Figure 5).

### 2.5. Overexpression of PtrMYB121 Activates the Expression of SCW Biosynthetic Genes

To better understand the molecular mechanisms of *PtrMYB121*-mediated SCW formation, especially the accumulation of lignin and cellulose, we used qRT-PCR analysis to analyze the expression profiles of SCW biosynthetic genes. From the results of qRT-PCR analysis, we found that all of the detected lignin synthetic genes (*4CL2*, *C3H1*, *CAD5*, *C4H*, *CCoAOMT1*, *COMT*, *F5H1,* and *PAL3*) had a significant increase in the *PtrMYB121*-overexpressing transgenic lines. Meanwhile, the expression of *4CL2*, *CCoAOMT1*, *COMT,* and *CAD5* was upregulated five- to 20-fold in transgenic lines (Figure 6). *CesA7* and *CesA8*, two cellulose biosynthetic genes, were upregulated five- to eight-fold in transgenic lines as compared with that in WT plants (Figure 6). We also tested the expression profiles of hemicellulose synthetic genes IRREGULAR XYLEM (*IRX*) 9 and *IRX*14, and they had no difference in WT and *PtrMYB121*-overexpressing transgenic lines (Figure 6). These results were consistent with the phenotypic observation and content determination of SCW composition in WT and *PtrMYB121*-overexpressing transgenic Arabidopsis (Figure 4 and Figure 5).

To confirm that *PtrMYB121* directly binds to the promoters of lignin and cellulose biosynthetic genes, we conducted the Cis-element analysis of the promoters of lignin (*4CL2*, *CCOAOMT1*, *COMT1,* and *CAD5*) and cellulose (*CesA7* and *CesA8*) biosynthetic genes. They all had secondary wall MYB-responsive elements ACC(A/T)A(A/C)(T/C) [39] in the promoter regions, and the number of elements were three (*4CL2*, *CAD5* and *CesA7*), four (*CCOAOMT1* and *COMT1*), and five *(CesA8*) (Appendix A). Then luciferase (LUC) activity assay was used in this experiment. The binary vector which contains a CaMV 35S promoter and the CDS sequence of *PtrMYB121* were used as the effector, while the vectors containing the six individual promoter fragments and LUC reporter genes were used as reporters (Figure 7A). Co-expression of effector and reporter in the leaves of *Nicotiana benthamiana* (*N. benthamiana*) significantly enhanced the relative LUC activity as compared with the mock (Figure 7B). The results suggest that *PtrMYB121* can bind to the promoters of lignin biosynthetic genes (*4CL2*, *CCoAOMT1*, *COMT*, and *CAD5*) and cellulose biosynthetic genes (*CesA7* and *CesA8*).

## 3. Discussion

SCW formation is a very complex process that requires coordinated regulation of a series of structural genes and TFs in vascular plants [40]. Researchers have identified various structural genes in SCW formation, including *4CL*, *C3H*, *CAD*, *C4H*, *CCOAOMT*, *COMT*, *F5H*, *PAL*, *CesA7*, *CesA8*, *IRX9,* and *IRX14* [41,42,43,44]. The transcription regulation of these genes is mainly controlled by the NAC- and MYB-mediated regulatory network. Some MYB TFs have been identified as specific activators of SCW formation in Arabidopsis. Among them, *AtMYB58*, *AtMYB63,* and *AtMYB85* can specifically activate the expression of lignin biosynthetic genes [45,46]. Meanwhile, *AtMYB61* plays crucial roles in lignin biosynthesis, trichome formation, and seed coat pigmentation [26,27,28,29,30]. Correspondingly, *AtMYB4*, *AtMYB32,* and *AtMYB7* have been proved to play negative regulatory roles in the biosynthesis of lignin and other phenylpropanoid compounds [47]. R2R3-MYB TFs were also identified to regulate the biosynthesis of SCW in other vascular plants, such as chrysanthemum, rice, maize, switchgrass, and eucalyptus [48,49,50]. In addition to Arabidopsis, the R2R3-MYB TFs are also involved in the regulation of SCW biosynthesis in other herbs. For example, overexpression of *SbMyb60* not only impacts phenylpropanoid biosynthesis, but also alters SCW composition in *Sorghum bicolor* [51]. In rice, *OsMYB103L* plays an important role in GA-mediated SCW synthesis, and functions as a master switch of downstream TFs [52]. *BdSWAM1* is a positive regulator of SCW thickening in *Brachypodium distachyon* and interacts with cellulose and lignin gene promoters [53].

As perennial plants, the SCW formation of woody plants is complicated, and its transcriptional regulation mechanism is largely unknown. There are at least 194 R2R3-MYB TFs in the poplar genome, and only a few of them have been demonstrated to regulate SCW formation. Meanwhile, *PtrMYB2/3*, *PtrMYB20/21*, and *PtrMYB74* are the MYB master genes that simultaneously regulate the synthesis of cellulose, lignin, and hemicellulose [50,54]; overexpression of *PtrMYB152* and *PtoMYB170/216* specifically promote the accumulation of lignin [35,36]. *PtoMYB92*, the homologous gene of *AtMYB85*, promotes the accumulation of lignin, but inhibits the hemicellulose synthesis [55]. In addition to activators, transcription suppressors *PtoMYB156* and *PdMYB221* were also found to inhibit the accumulation of cellulose, lignin, and hemicellulose [56,57]. Recently, some other TFs have also been shown to participate in the process of SCW biosynthesis in addition to R2R3-MYB TFs. The overexpression of a pine Dof TF *PpDof5* in hybrid poplars promoted the accumulation of lignin in stems and basal leaves [58]. The *ARK1* gene from the ARBORKNOX family reduced the lignin content by downregulating lignin synthesis-related genes [59]. KNAT 2/6b, one member of the class I KNOX genes, inhibited the xylem differentiation by regulating NAC TF in poplar [60]. The overexpression of *PdOLP1*, an osmotin-like protein gene in poplar, caused a reduction in the radial width and cell layer number in xylem and phloem zones, and downregulated expression of genes involved in lignin biosynthesis, and in the fibers and vessels of xylem cell walls [61].

In this study, we characterized and isolated a R2R3-MYB TF *PtrMYB121* from *P. trichocarpa* and demonstrated that *PtrMYB121* positively regulated the biosynthesis of lignin and cellulose. Phylogenetic analysis showed that *PtrMYB55/121* and *PtoMYB170/216* from *P. tomentosa* were homologous genes of Arabidopsis *AtMYB55* and *AtMYB61* (Figure 1A). Among them, *AtMYB61* and *PtoMYB170/216* have been proved to play key roles in SCW synthesis [26,27,28,29,30,35,36]. Amino acid sequence alignment showed that the N-terminal region of *PtrMYB55/121* contain conserved R2 and R3 domains with their homologous genes, and differ in the C-terminal region (Figure 1B). Therefore, we speculate that *PtrMYB55/121* are also involved in SCW biosynthesis but have functional differentiation with their homologous genes. Tissue expression pattern analysis showed that *PtrMYB55* and *PtrMYB121* had the same expression patterns, and were preferentially in the old stems, especially in vascular tissues and organs (Figure 2). The results indicated that *PtrMYB55* and *PtrMYB121* should regulate SCW formation and had functional redundancy in the process. Meanwhile, the expression profiles of *PtrMYB121* in all detected tissues were higher than *PtrMYB55* (Figure 2). Therefore, *PtrMYB121* was selected for further research. Overexpression of *PtrMYB121* caused the wider xylem and ectopic deposition of lignin (Figure 4), which were consistent with the measurement of lignin content (Figure 5). Interestingly, the cellulose content in *PtrMYB121*-overexpressing transgenic lines were also significantly increased, and the content of hemicellulose had no change (Figure 5). *PtoMYB170* positively regulated the lignin deposition and conferred drought tolerance [35], while *PtoMYB216* only promoted the accumulation of lignin [36]. The results were different from *PtoMYB170/216* [35,36], indicating that *PtrMYB121* had functional differentiation with its homologous genes.

In general, NAC- and MYB-mediated SCW biosynthesis was regulated by the downstream structural genes [62]. The qRT-PCR analysis was used to analyze the expression level of SCW biosynthetic structural genes and showed that the expression of lignin and cellulose biosynthetic genes was significantly upregulated in *PtrMYB121*-overexpressing lines but the expression of hemicellulose biosynthetic genes had no change (Figure 6). The results were consistent with phenotype observation and measurement of SCW components and the LUC activity assay confirmed that *PtrMYB121* could bind to the promoters of lignin and cellulose biosynthetic genes and activate their expression (Figure 7).

On the basis of our results combined with the results of previous studies, we can conclude that *PtrMYB55/121* and *PtoMYB170/216* all positively regulate the accumulation of lignin. *PtrMYB121* also promotes cellulose biosynthesis, which is the functional differentiation with *PtoMYB170/216* [35,36]. It is worth mentioning that *AtMYB61* also affects the lateral root development and color pigmentation, however, whether or not *PtrMYB121* has similar functions remains to be further studied.

In conclusion, this study confirmed that *PtrMYB121* could bind to the lignin and cellulose biosynthetic genes to activate their expression, thereby promoting the accumulation of lignin and cellulose. This study provides some information about the complex regulatory network of SCW biosynthesis and could help to design strategies for wood genetic improvement.

## 4. Materials and Methods

### 4.1. Plant Materials

The seedlings of *P. tricocharpa* and seeds of *A. thaliana* (Columbia) and *N. benthamiana* were used in this study. Seeds of *A. thaliana* and *N. benthamiana* were sterilized with 70% ethanol (30 s) and 1% sodium hypochlorite (10 min), and then germinated on plates with MS medium (Hope Bio, Qingdao, China) supplemented with 8 g/L agar. Fourteen days later, the seedings were transferred into soil and grown in a greenhouse. The culture conditions of *P. trichocarpa* and *N. benthamiana* were 16/8 h light/dark cycle, 23–25 °C, 10,000 Lux light, and relative humidity of about 40%. The culture conditions of Arabidopsis were 16/8 h light/dark cycle, 22–23 °C, 5000 Lux light, and relative humidity of about 60% [63].

### 4.2. Phylogenetic Analysis

The amino acid sequences of R2R3-MYB TFs were obtained from the GenBank of NCBI (https://www.ncbi.nlm.nih.gov). Then, the sequences were used to construct the phylogenetic tree by neighbor-joining (NJ) method in p-distance model using MEGA8.0 software (Park, Pennsylvania, USA) [64] (https://www.megasoftware.net/), and multiple sequence alignment was carried out with DNAMAN software [65].

### 4.3. RNA Isolation and qRT-PCR Analysis

Total RNA was extracted from various tissues (young leaf, old leaf, young stem, old stem, and root) of *P. trichocarpa* and *A. thaliana* with the Trizol Reagent (QIAGEN), following the manufacturer’s instructions. Then, the reverse-transcription PCR was performed with the PrimeScript RT reagent Kit (QIAGEN) to synthesize the first-strand cDNA. The cDNA was stored at −20 °C and used as the template for gene cloning and qRT-PCR. The qRT-PCR analysis was performed as previous studies [63].

### 4.4. Cloning of Full-Length PtrMYB121 Coding Sequence (CDS)

The full-length CDS (without the termination codon) of *PtrMYB121* was amplified by PCR reaction with gene-specific primers listed in Appendix A. The PCR products were cloned into the plant binary vector pCAMBIA1305 containing a CaMV 35S promoter. The 35S: *PtrMYB121* binary vector was introduced into the *Agrobacterium tumefaciens* strains GV3101. The reaction system (25 μL) contained 12.5 μL Prime STAR DNA polymerase (Takara, Kyoto, Japan), 0.25 μM forward primer, 0.25 μM reverse primer, 100 ng DNA template from stems of *P. trichocarpa,* and supplemented with ddH2O. The procedure was as follows: 94 °C 300 s; 94 °C 45 s, 58 °C 60 s, and 72 °C 90 s for 30 cycles; 72 °C 600 s.

### 4.5. Arabidopsis Transformation

The 35S: *PtrMYB121* binary vector was transformed into 30-day-old Arabidopsis plants by floral dip method [66]. Transgenic plants were selected on 1/2 MS medium containing 50 mg/L hygromycin and 400 mg/L cephalosporin. The survived seedlings were transferred into the soil and grown in the greenhouse. For these seedlings (T1) in greenhouse, we individually collected their seeds (T1) for every line for further culture. The seeds of each line were sterilized with 75% ethanol (30 s), 1.5% sodium hypochlorite (10 min), and further selected on 1/2 MS medium containing 50 mg/L hygromycin and 400 mg/L cephalosporin. The lines (T2) with the separation ratio (survival/death = 3:1) were used as single copy homozygotes. The positive transgenic Arabidopsis were characterized by PCR and qRT-PCR analysis for further research.

### 4.6. Phenotypic Observation of PtrMYB121-Overexpressing Lines in Arabidopsis

TBO staining was used to observe the lignin deposition. The bases of inflorescence stems of 40-day-old Arabidopsis were sliced, and paraffin sections (15 mm) were made with an ultra-thin semiautomatic microtome (FINESSE 325, Thermo, Waltham, USA), as previous studies [63]. Hence, the paraffin sections were stained with 1.0% (*W*/*V*) TBO for 30 s immediately, and the temporary slides were observed with an optical microscope. Among them, the location of blue pigmentation represented the xylem.

Paraffin sections were also used for the observation of lignin autofluorescence, and the temporary slides were used for the detection of fluorescence signal using a confocal laser scanning microscopy (Leica TCS SP8 X, Germany) at 408 nm laser.

### 4.7. Measurement of Lignin, Cellulose, and Hemicellulose Components in Arabidopsis

The components of SCW were measured, as previously described [55]. The acetyl bromide (AcBr) method was used to estimate the total lignin content [37]. The cellulose and hemicellulose content were measured by the Van Soest method [38].

### 4.8. LUC Activity Assay

The full-length coding sequence (CDS) of *PtrMYB121* was amplified by PCR amplification with gene-specific primers (Appendix A) and ligated into the plant binary vector pCAMBIA1302 driven by the CaMV 35S promoter; the construct was used as effector for the LUC activity assay. The promoter fragments (*4CL2* 1656bp, *CAD5* 1733bp, *CCoAOMT1* 1687bp, *COMT* 1673 bp, *CesA7* 1466bp, and *CesA8* 1784 bp) were independently cloned (Appendix A) and ligated into the pGreen-0800-35mini vector to produce various LUS reporters; these constructs were used as effectors. All these vectors were individually transformed into the *A. tumefaciens* strains GV3101. The reporters and effector were simultaneously infiltrated into the leaf epidermal cells of *N. benthamiana,* as described above [67]. After 48 h incubation, the LUC activity was measured with a GloMax^®^ 20/20 luminometer.

### 4.9. Statistical Analysis

All experimental data were obtained from three biological replicates, and the statistical analysis were performed with Student’s *t*-test. In all experiments, significant differences of the data were evaluated by one-way ANOVA. * *p* < 0.05 and ** *p* < 0.01.

### 4.10. GenBank Accession Numbers of Genes Used in this Study

The GenBank accession numbers of other genes used in this study are as follows: PtrMYB121 (XP_002302702.1), AtMYB46 (AT5G12870.1), AtMYB55 (NM_116398.4), AtMYB58 (AF062893.1), AtMYB61 (AF062896.1), AtMYB63 (AT1G79180), AtMYB83 (AT3G08500.1), AtMYB103 (AT1G63910.1), EgMYB1 (AJ576024), EgMYB2 (AJ576023.1), HvMYB3 (X70881.1), OsMYB46 (JN634085), PtMYB1 (AY356372.1), PtMYB4 (AY356371.1), PtMYB8 (DQ399057.1), PtoMYB74 (KX887329.1), PtoMYB92 (KP710214.1), PtoMYB170 (KY114929.1), PtoMYB216 (JQ801749.1), PtrMYB2 (KF148677.1), PtrMYB3 (KF148675.1), PtrMYB20 (KF148676.1), PtrMYB55 (XM_002302666.3), PtrMYB192 (XM_002310643.2), PvMYB4a (JF299185), ZmMYB31 (NM_001112479), ZmMYB42 (NM_001112539)

## Figures and Tables

**Figure 1 ijms-21-07734-f001:**
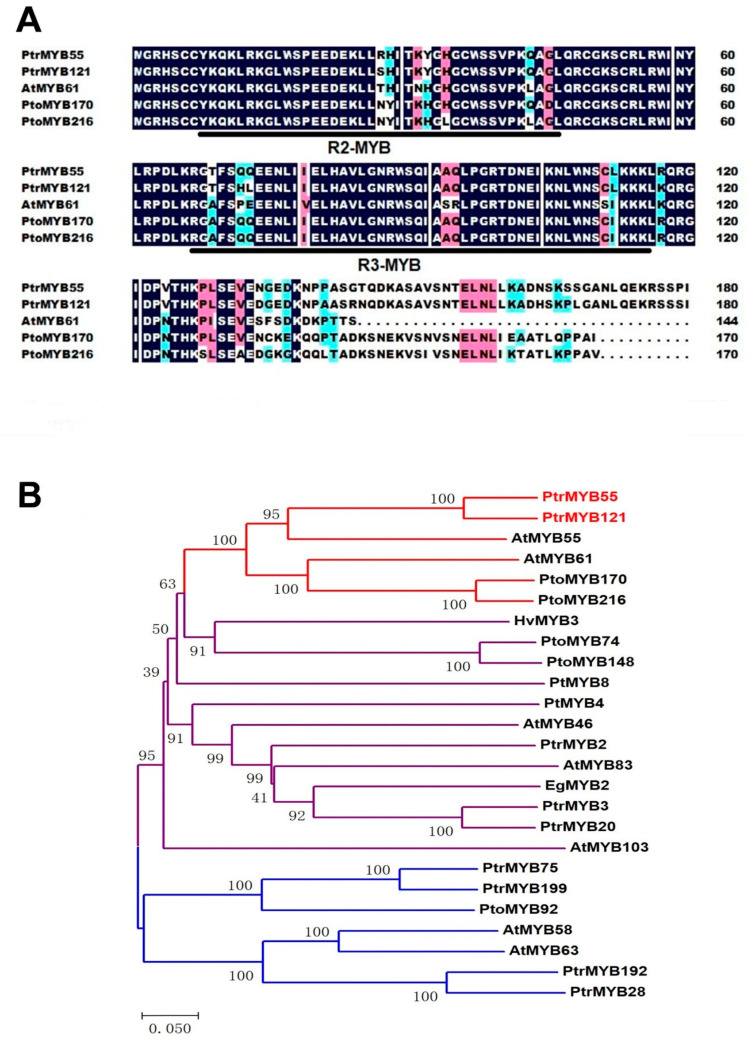
Sequence alignment and phylogenetic analysis of *PtrMYB55* and *PtrMYB121*. (**A**) Amino acid sequence alignment of the N-terminal region of *PtrMYB55* and *PtrMYB121* (*P. trichocarpa*) with *AtMYB61* (*A. thaliana*), *PtoMYB170* (*Populus tomentosa*), and *PtoMYB216* (*Populus tomentosa*). The amino acid sequences were aligned with DNAMAN software. The conserved R2 and R3 domains are underlined. Blue and pink shades are used to indicate the similarity of amino acid residues, which is 75–100% and 50–75%, respectively; (**B**) Phylogenetic analysis of *PtrMYB55* and *PtrMYB121* with R2R3-MYB TFs involved in secondary cell wall (SCW) formation in vascular plants. Phylogenetic tree was analyzed with MEGA8.0. Bar = 0.050 substitutions per site. *PtrMYB55* and *PtrMYB121* and its homologous genes are highlighted with red lines. The GenBank accession numbers of these R2R3-MYB TFs are listed in Materials and Methods.

**Figure 2 ijms-21-07734-f002:**
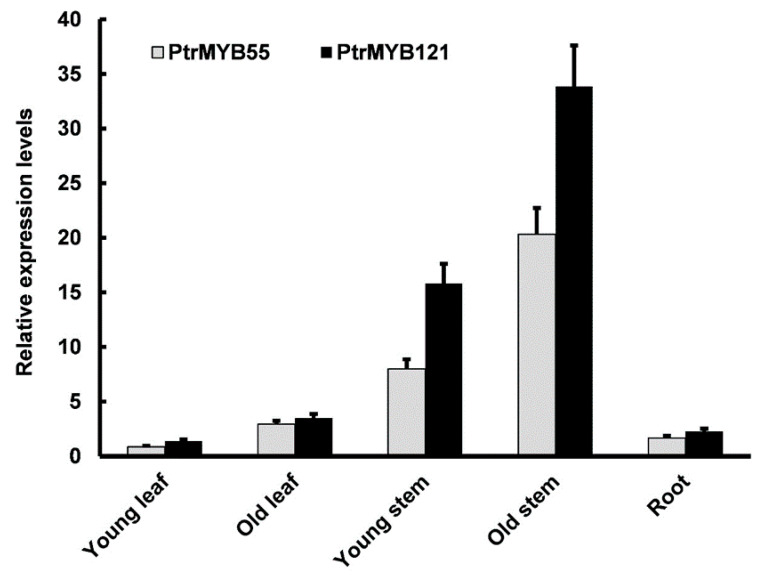
Tissue expression patterns of *PtrMYB55* and *PtrMYB121* in *P. trichocarpa* seedlings. qRT-PCR was used to analyze the relative expression level of *PtrMYB55* and *PtrMYB121* in different tissues of wild-type poplar. The poplar *Actin* gene was used as internal control. Error bars represent ± SD from three biological repeats.

**Figure 3 ijms-21-07734-f003:**
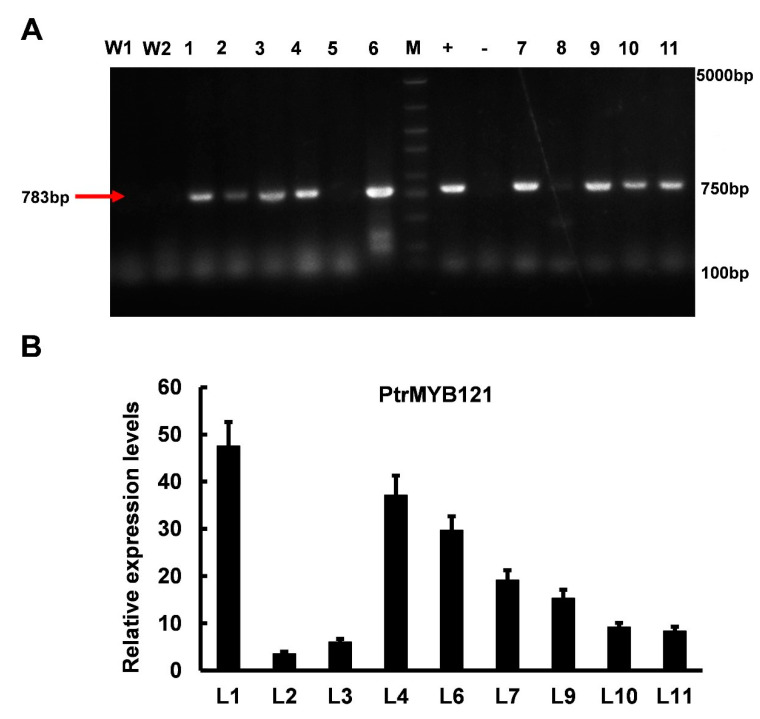
Different transgenic lines of *PtrMYB121*-overexpressing Arabidopsis. (**A**) PCR products of different *PtrMYB121*-overexpressing lines. W1, wild-type line 1; W2, wild-type line 2; +, positive control (plasmid used for transformation containing hygromycin resistance gene); −, negative control (ddH_2_O); (**B**) Expression profiles of *PtrMYB121* in the inflorescence stems of different transgenic lines of T2 generation Arabidopsis.

**Figure 4 ijms-21-07734-f004:**
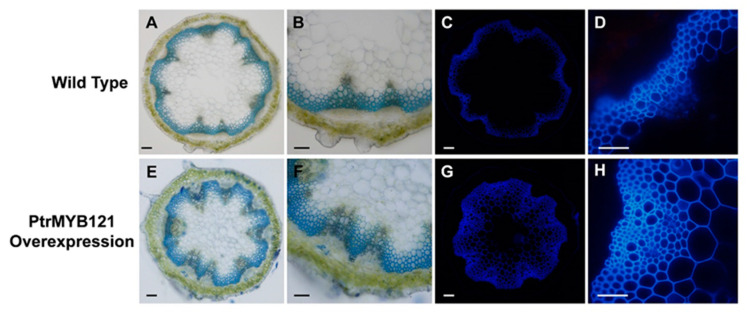
Tissue sections of inflorescence stems of *PtrMYB121*-overexpressing transgenic Arabidopsis. The toluidine blue O (TBO) staining and lignin autofluorescence in the inflorescence stems of 40-day-old WT (WT-1) and *PtrMYB121*-overexpressing Arabidopsis (L1). Histochemical staining of the cross-sections of inflorescence stem of 40-day-old WT (**A**,**B**) and *PtrMYB121*-overexpressing (**E**,**F**) Arabidopsis. Lignin autofluorescence of the inflorescence stem of 40-day-old WT (**C**,**D**) and *PtrMYB121*-overexpressing (**G**,**H**) Arabidopsis. Bars: A,C,E,G = 500 µm and B,D,F,H = 100 µm.

**Figure 5 ijms-21-07734-f005:**
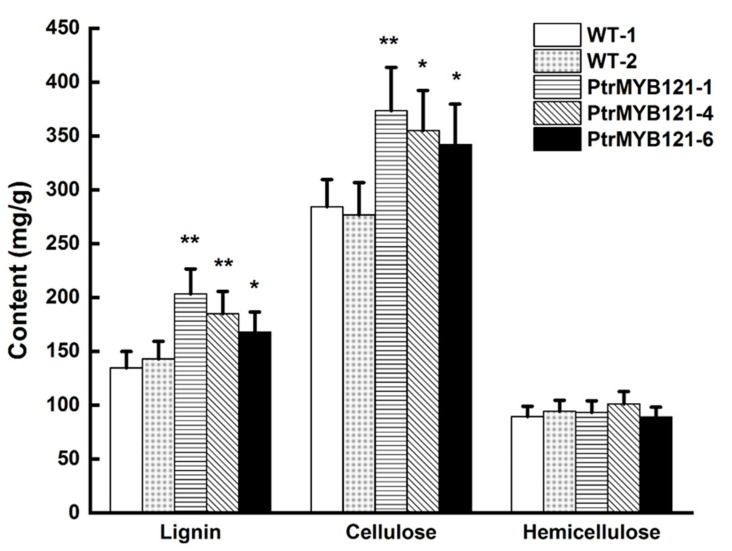
Content of SCW components (lignin, cellulose, and hemicellulose) in the inflorescence stems of WT and *PtrMYB121*-overexpressing transgenic Arabidopsis. Error bars represent ± SD from three biological repeats. Significant difference between different lines was tested by Student’ s *t*-test. * *p* < 0.05 and ** *p* < 0.01.

**Figure 6 ijms-21-07734-f006:**
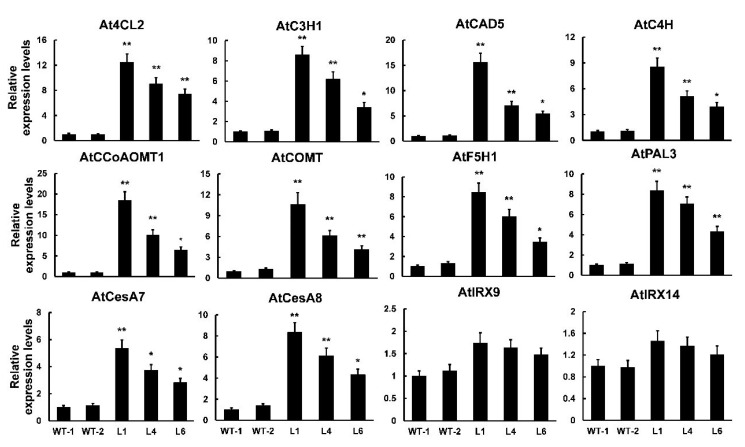
Expression profiles of SCW biosynthetic genes in WT and *PtrMYB121*-overexpressing plants. *4CL2*, *C3H1*, *CAD5*, *C4H*, *CCoAOMT1*, *COMT*, *F5H1,* and *PAL3*: lignin biosynthetic genes. *CesA7* and *CesA8*: cellulose biosynthetic genes. *IRX9* and *IRX14*: hemicellulose biosynthetic genes. Gene expression profiles were evaluated using the 2^−∆∆*Ct*^ method, and the expression of relative genes in WT-l was set as 1. Error bars represent ± SD from three biological repeats. Significant difference between different lines was tested by Student’ s *t*-test. * *p* < 0.05 and ** *p* < 0.01.

**Figure 7 ijms-21-07734-f007:**
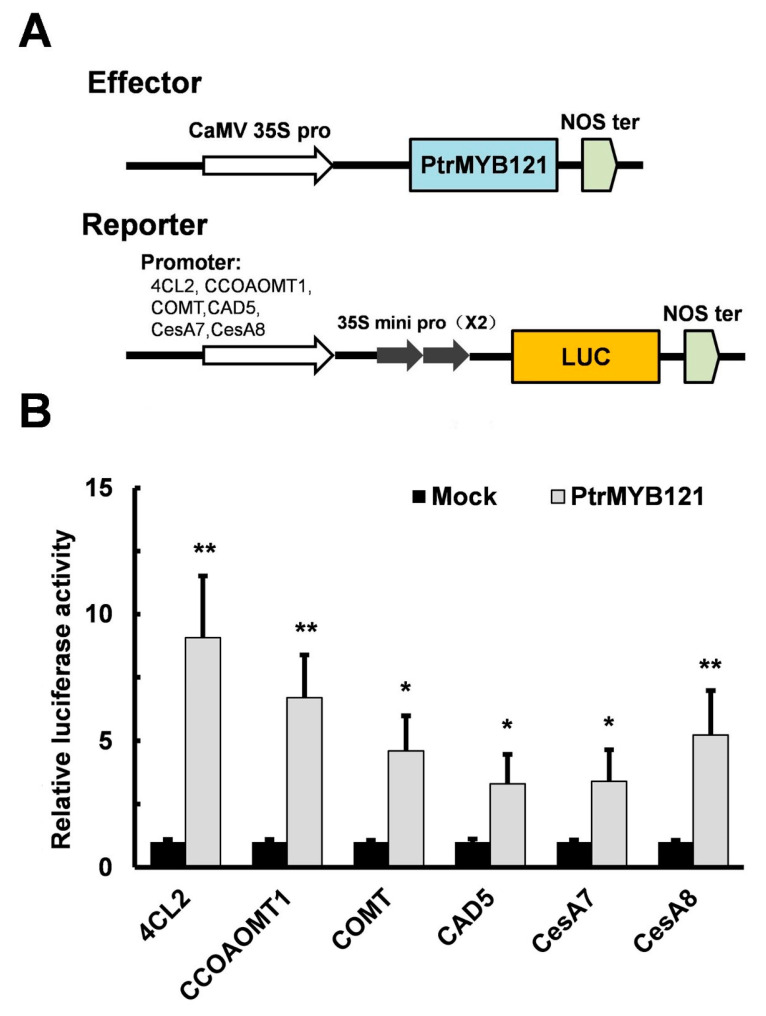
*PtrMYB121* activates the expression of lignin and cellulose biosynthetic genes. (**A**) Pattern diagrams of the construction vectors in the effector-reporter system for transient activation analysis. The effector is the coding sequence (CDS) of *PtrMYB121* driven by a 35S promoter. The reporters contain the luciferase (LUC) reporter gene driven by the 6 individual promoters of Arabidopsis *4CL2*, *CCoAOMT1*, *COMT*, *CAD5*, *CesA7,* and *CesA8* genes, respectively; (**B**) Transient activation analysis shows that *PtrMYB121* can promote the expression of reporters. All these effectors were individually transformed into the leaves of *N. benthamiana* as the mock treatments, and the relative expression of their LUC activity was set to 1. Error bars represent ± SD from three biological repeats. Student’s *t*-test: * *p* < 0.05 and ** *p* < 0.01.

**Table 1 ijms-21-07734-t001:** Cell layer number and radial width of xylem in inflorescence stems of wild-type (WT-1) and *PtrMYB121*-overexpressing transgenic (L1) Arabidopsis plants.

Line	Samples	Cell Layer (n)	Radial Width (μm)
WT-1	1	6.78 ± 0.81 ^a^	439.72 ± 28.79 ^a^
	2	7.11 ± 0.94 ^a^	461.77 ± 33.47 ^a^
	3	6.43 ± 0.87 ^a^	422.36 ± 30.16 ^a^
PtrMYB121-L1	1	13.76 ± 1.57 ^b^	657.42 ± 53.34 ^b^
	2	14.20 ± 1.72 ^b^	693.17 ± 58.38 ^b^
	3	13.49 ± 1.55 ^b^	638.84 ± 49.67 ^b^

Different letters (a, b) above bars represent statistically significant differences between groups (*p* < 0.05) as determined by one-way ANOVA followed by Dunnett’s test. The data with same letter (a or b) meant that they had no significant difference between groups, whereas the different letters (a and b) represented significant differences between groups.

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
