# Peer review of "Overexpression of PtrMYB121 Positively Regulates the Formation of Secondary Cell Wall in Arabidopsis thaliana"

_ijms, 2020, doi:10.3390/ijms21207734_

Round 1

Reviewer 1 Report

The manuscript, Ying Liu  et. al (I.D.:ijms-956054) describes the overexpression of PtrMYB121 gene in Arabidopsis, and study its function in cell wall biosynthesis.

I have some comments and suggestion for the Authors.

Line 6: Please check and correct the author’s name

Line 33: Please use a full name for the SCW too. During the first appearance in the article, you have to write the full name of the expression instead of its abbreviation.

Line 98: It is not clear for the reviewer why the authors used the AtMYB61 sequence for homology search? In the introduction part there is no any information about this gene. Please add information about this gene to the introduction part.

Line 121: The following abbreviation is deceptive: PtrMYB121/55 Please use rather PtrMYB121 and PtrMYB55. And please check it through the article.

Line 127: Please use just simple “low expression” wording instead of “extremely low”.

Line138: Please rewrite this sentence, and remove the “binary vector”.

Line 139:  Why did the authors use the T2 generation? I presume that the authors used a homozygous line. If yes, please rewrite this sentence and add “T2 homozygous line”

line 140: Please rewrite this sentence. Not the CDS sequence, but just a 738bp long fragment was amplified.

Line 142: Please remove the expression of “independent lines”. The qRT-PCR is unsuitable for determining the independent lines.

Line 143: What does “for functional verification” expression mean? Please specify or rewrite this sentence. The qRT-PCR is unsuitable for the “functional verification”

Line 148 “recombinant plasmid….” please rewrite this sentence, for example use this:“plasmid used for transformation”

Line 158: How do the Authors perform the calculation of the percentage? Please add a short description about this.

Line 159: Please check the figure legend usage: E and F are not the result of a wild type plant.

Line 162: as line 159. Please correct it.

Line 183: please remove the word “research”, it is unnecessary.

Line 185: “lignin synthetic genes” During the first appearance in the article, you have to write the full name of the expression instead of its abbreviation.

Line 188: same as Line 185. And please check it through the article.

Line 195: Figure 6. The authors wrote: “Relative expression”. What does it mean? How did the authors perform the calculation?

Line 202: Authors wrote: “biosynthetic structural genes”. What do they mean? Please rewrite this sentence.  

Line 213 and Figure 7. panel A: It is not clearly presented, whether the Authors constructed only one reporter construct with 7 promoters, or 7 individual constructs were created. Please clarify this.

Line 225-228: Move this part to the Introduction part, where the possible function of AtMYB61 is discussed.

Line282: Please specify the reference company, composition, sugar content, etc. of the MS medium.

Line 290: Please add reference for MEGA8.0. (Web page or article)

Line 293: What do “relative tissues” mean? Remove or specify them.

Line 299: Why did the Authors remove the STOP codon? Did they generate a fusion protein? If they did not, the translation will not stop, and you will get an extra unknow protein sequence on your protein of interest.

Line 299-300: Please specify the condition of PCR reaction. Enzyme, methods, company etc.

Line 307: How did the authors perform the “screening for single copy homozygotes” ?

Line 309: Please add more detailed information about this method.

Line 325: Wrong reference “56” Please correct it.

Author Response

Line 6: Please check and correct the author’s name

 Thank you for your suggestion. we have checked the author’s name, and they are right.

Line 33: Please use a full name for the SCW too. During the first appearance in the article, you have to write the full name of the expression instead of its abbreviation.

  Thank you for your suggestion. We have changed it, and wrote the full name of SCW (secondary cell wall)

Line 98: It is not clear for the reviewer why the authors used the AtMYB61 sequence for homology search? In the introduction part there is no any information about this gene. Please add information about this gene to the introduction part.

 Yes, thank you for your comments. we forgot to describe the importance of AtMYB61. We have added the description of AtMYB61in the introduction part “Meanwhile, the AtMYB61 was founded to be a multifunctional TF in Arabidopsis. It can specifically regulate lignin biosynthesis, affect the trichome formation and root development, regulate the stomata development, also have effects on the coloring of seed coat. Furthermore, AtMYB61 can also link the transcription regulation of multiple aspects of resource allocation.”

Line 121: The following abbreviation is deceptive: PtrMYB121/55 Please use rather PtrMYB121 and PtrMYB55. And please check it through the article.

 Thank you for your suggestion. We have changed the deceptive abbreviation and checked it through the article.

Line 127: Please use just simple “low expression” wording instead of “extremely low”.

  Thank you for your comments. We have replaced  the words “extremely low” with “low expression”.

Line138: Please rewrite this sentence, and remove the “binary vector”.

 Thank you for your comments. We have removed the “binary vector” and rewritten this sentence “To analyze the function of PtrMYB121 in SCW biosynthesis, we overexpressed its full-length coding sequence (CDS) in wild-type A. thaliana.”

Line 139:  Why did the authors use the T2 generation? I presume that the authors used a homozygous line. If yes, please rewrite this sentence and add “T2 homozygous line”

 Yes, we wanted to showed that we used the homozygous line in our research. We have rewritten this sentence and add “T2 homozygous line”. “Then, we obtained 11 hygromycin-resistant A. thaliana lines and analyzed the function of PtrMYB121 in the T2 homozygous generation.”

line 140: Please rewrite this sentence. Not the CDS sequence, but just a 738bp long fragment was amplified.

 Thank you for your suggestion. We have rewritten the sentence “A part (783bp) of the CDS sequence of PtrMYB121 was amplified by PCR using gene-specific primers”.

Line 142: Please remove the expression of “independent lines”. The qRT-PCR is unsuitable for determining the independent lines.

 Thank you for your suggestion. We have removed the “independent lines”

Line 143: What does “for functional verification” expression mean? Please specify or rewrite this sentence. The qRT-PCR is unsuitable for the “functional verification”

  I apologize for causing your misunderstanding. The qRT-PCR is unsuitable for the “functional verification”. In this process, we used qRT-PCR analysis to analyze the relative expression of PtrMYB121 and select the lines for further functional verification. We have rewritten this sentence “qRT-PCR analysis was used to select candidate lines of PtrMYB121-overexpressing transgenic Arabidopsis for later functional verification”

Line 148 “recombinant plasmid….” please rewrite this sentence, for example use this:“plasmid used for transformation”

  Thank you for your suggestion. We have rewritten the sentence with “plasmid used for transformation”

Line 158: How do the Authors perform the calculation of the percentage? Please add a short description about this.

 We have added a description for the calculation of the percentage of xylem area. “To quantitatively analyze the changes in xylem of PtrMYB121-overexpressing transgenic Arabidopsis, we made the anatomical cross-sections of inflorescence stems and took pictures with an optical microscope system. Then the Image J software was used for calculating the width of xylem area, and every samples had 30 slices for analysis.”

Line 159: Please check the figure legend usage: E and F are not the result of a wild type plant.

 Thank you, it is our mistake. We have changed it

Line 162: as line 159. Please correct it.

  Thank you, it is our mistake. We have changed it.

Line 183: please remove the word “research”, it is unnecessary.

 Thank you, it is our mistake. We have removed the word “research”.

Line 185: “lignin synthetic genes” During the first appearance in the article, you have to write the full name of the expression instead of its abbreviation.

  Thank you for your attention. We have described their full name in the part of “Introduction”. 4CL2 (4-coumarate-CoA ligase 2), C3H1(p-coumarate 3-hydroxylase 1), CAD5 (cinnamyl alcohol dehydrogenase 5), C4H (cinnamate 4-hydroxylase), CCoAOMT1 (CoA3-O-methyltransferase 1), COMT (caffeic acid O-methyltransferase), F5H1(ferulate 5-hydroxylase1) and PAL3 (phenylalanine ammonialyase 3).

Line 188: same as Line 185. And please check it through the article.

 Thank you for your attention. We have described their full name in the part of “Introduction”. CesA (cellulose synthase). We have also checked it through the article and added the full time when they are the first appearance.

Line 195: Figure 6. The authors wrote: “Relative expression”. What does it mean? How did the authors perform the calculation?

 Thank you for your comments. We have added the performance of qRT-PCR. “Arabidopsis UBQ gene was used as internal control. Gene expression profiles were evaluated using the 2−∆∆Ct method, and the expression of relative genes in WT-l was set as 1.”

Line 202: Authors wrote: “biosynthetic structural genes”. What do they mean? Please rewrite this sentence.  

  Thank you, it is our mistake. We just wanted to mean the genes involved in the biosynthesis of lignin and cellulose. We have removed the word “structural”.

Line 213 and Figure 7. panel A: It is not clearly presented, whether the Authors constructed only one reporter construct with 7 promoters, or 7 individual constructs were created. Please clarify this.

  Thank you for your suggestion. The description for vector construction is not clearly presented. For reporters, 7 individual constructs were created. And we have added the detail information about LUC assay.

Line 225-228: Move this part to the Introduction part, where the possible function of AtMYB61 is discussed.

 Thank you for your suggestion. We have moved them into the Introduction part

Line282: Please specify the reference company, composition, sugar content, etc. of the MS medium.

 Thank you for your suggestion. We have added the information of MS medium.

Line 290: Please add reference for MEGA8.0. (Web page or article)

  Thank you for your suggestion. We have added the Web page and reference for MEGA8.0.

Line 293: What do “relative tissues” mean? Remove or specify them.

Thank you for your suggestion. We have added the description for relative tissues (young leaf, old leaf, young stem, old stem, and root).

Line 299: Why did the Authors remove the STOP codon? Did they generate a fusion protein? If they did not, the translation will not stop, and you will get an extra unknow protein sequence on your protein of interest.

 Thank you for your comments. As you assumed, we removed the STOP codon to generate a fusion protein with Green fluorescent protein (GFP) for overexpression of PtrMYB121, and subcellular location analysis in another research (unsubmitted)

Line 299-300: Please specify the condition of PCR reaction. Enzyme, methods, company etc.

 Thank you for your suggestion. We have added the information for PCR reaction “The reaction system (25 μL) contains 12.5 μL Prime STAR DNA polymerase (Takara, Kyoto, Japan), 0.25 μM forward primer, 0.25 μM reverse primer, 100 ng DNA template from stems of P. trichocarpa, and supplemented with ddH2O. The procedure is as follows: 94℃,300s; 94℃ 45s, 58℃ 60s, 72℃ 90s,30 cyclesï¼› 72℃,600s.”

Line 307: How did the authors perform the “screening for single copy homozygotes” ?

Thank you for your suggestion. We have added the information about screening for single copy homozygotes “Transgenic plants were selected on 1/2 MS medium containing 50 mg/L hygromycin and 400 mg/L cephalosporin. The survived seedlings were transferred into the soil and grown in the greenhouse. For these seedlings(T1) in greenhouse, we individually collected their seeds (T1) of every line for further culture. The seeds of each line were sterilized with 75% Ethanol (30s), 1.5% sodium hypochlorite (10 min), and further selected on 1/2 MS medium containing 50 mg/L hygromycin and 400 mg/L cephalosporin. The lines(T2) with the separation ratio (survival: death = 3:1) can be used as single copy homozygotes.”

Line 309: Please add more detailed information about this method.

 Thank you for your suggestion. We have added the information about TBO staining.“TBO staining was used to observe the lignin deposition. The bases of inflorescence stems of 40-day-old Arabidopsis were sliced, and paraffin sections (15 mm) were made with an ultra-thin semi-automatic microtome (FINESSE 325, Thermo, Waltham, USA) as previous studies. Hence, the paraffin sections were stained with 1.0% (W/V) TBO for 30 s immediately, and the temporary slides were observed with an optical microscope. Among them, the location of blue pigmentation is the xylem.”

Line 325: Wrong reference “56” Please correct it.

Thank you for your suggestion. We have corrected it.

Reviewer 2 Report

The manuscript describes functional study of one transcription factor from Myb family and its role in secondary cell wall formation, especially lignin biosynthesis. In overall, the experimental design is clear and quite simple. On the other hand, some analyses could be performed more in detail (will be explained later). The studied gene was overexpressed just in heterologous plant, i.e. Arabidopsis thaliana, not a woody plant. The authors should briefly state why they did not overexpress the gene in poplar (time-consuming, low efficiency, possible silencing effect etc…).

The introductory part contains comprehensive overview of the secondary cell wall composition, biosynthesis and its regulation. In row 86 the authors mention 194 Mybs identified in poplar (by the way, the readers should know clearly if they describe Populus trichocarpa, if another species or just Populus sp., not just Ptr or Pto…) and less than 10% is mentioned to be functionally studied. At this point references are missing. The PtrMyb121 gene was successfully overexpressed in Arabidopsis and 11 hygromycin resistant plants were obtained. But in Material and Methods the authors mention that they scored the transgenic lines for number of T-DNA inserts. It is not clear, whether these 11, or 9 (PCR confirmed) lines contain just one insert or more.

The luciferase assay showing activation of selected promoters by PtrMyb121 gene could be described more in detail. The reader has no information about length of promoters, reporter constructs, analysis of cis acting elements, etc. Possible DNA binding site for PtrMyb121 could be outlined. Picture showing Myb binding sites in the studied promoters could be nice part of the Supplementary material and probably some consequence in promoters composition of lignin, cellulose and hemicellulose genes could be found.

So my conclusion is to accept the manuscript after detailed analysis of binding sites or promoter analyses and improving the discussion part. Some other examples of effect of overexpression of SCW component biosynthetic genes should be given.

Other comments:

  • Did the authors check also trichome density? (Even if there was no significant difference in phenotypes of wt and transgenic plants, some table comparing different parameters could be shown in Supplementary material)
  • 50mg/l hygromycin is quite a high dose for Arabidopsis, usually 15-20 is sufficient.
  • English should be checked, especially verbs.
  • Avoid using abbreviances without explaining at least at once (SCW etc…)
  • Figs: error bars seem to be the same or very regular (could be possible, but I have another experiences from my own qPCR data)

Author Response

The manuscript describes functional study of one transcription factor from Myb family and its role in secondary cell wall formation, especially lignin biosynthesis. In overall, the experimental design is clear and quite simple. On the other hand, some analyses could be performed more in detail (will be explained later). The studied gene was overexpressed just in heterologous plant, i.e. Arabidopsis thaliana, not a woody plant. The authors should briefly state why they did not overexpress the gene in poplar (time-consuming, low efficiency, possible silencing effect etc…).

Thank you for your comments. The best strategy for this paper is overexpressing the PtrMYB121 in poplar. However, our lab has not established a mature genetic transformation system in poplar when we started the research. At that time, it took 10 months to get transgenic lines, and the efficiency was also very low (<5%). Now, we can get transgenic lines within 4 months, and the transformation efficiency is nearly 40%. Genetically modified poplar will be introduced in our future research.

The introductory part contains comprehensive overview of the secondary cell wall composition, biosynthesis and its regulation. In row 86 the authors mention 194 Mybs identified in poplar (by the way, the readers should know clearly if they describe Populus trichocarpa, if another species or just Populus sp., not just Ptr or Pto…) and less than 10% is mentioned to be functionally studied. At this point references are missing. The PtrMyb121 gene was successfully overexpressed in Arabidopsis and 11 hygromycin resistant plants were obtained. But in Material and Methods the authors mention that they scored the transgenic lines for number of T-DNA inserts. It is not clear, whether these 11, or 9 (PCR confirmed) lines contain just one insert or more.

Thank you for your comments. It is our mistake. 194 R2R3-MYB transcription factors were identified in Populus trichocarpa. We have rephrased the sentence and added the reference. The transgenic lines contain just one T-DNA insert. We have added the description “Transgenic plants were selected on 1/2 MS medium containing 50 mg/L hygromycin and 400 mg/L cephalosporin. The survived seedlings were transferred into the soil and grown in the greenhouse. For these seedlings(T1) in greenhouse, we individually collected their seeds (T1) of every line for further culture. The seeds of each line were sterilized with 75% Ethanol (30s), 1.5% sodium hypochlorite (10 min), and further selected on 1/2 MS medium containing 50 mg/L hygromycin and 400 mg/L cephalosporin. The lines(T2) with the separation ratio (survival: death = 3:1) can be used as single copy homozygotes.”

The luciferase assay showing activation of selected promoters by PtrMyb121 gene could be described more in detail. The reader has no information about length of promoters, reporter constructs, analysis of cis acting elements, etc. Possible DNA binding site for PtrMyb121 could be outlined. Picture showing Myb binding sites in the studied promoters could be nice part of the Supplementary material and probably some consequence in promoters composition of lignin, cellulose and hemicellulose genes could be found.

Thank you for your comments. We have added the detail information of the luciferase assay in “Materials and Methods” and “Results”. “we made the Cis-element analysis of the promoters of lignin (4CL2, CCOAOMT1, COMT1 and CAD5) and cellulose (CesA7 and CesA8) biosynthetic genes. They all had the secondary wall MYB-responsive elements ACC(A/T)A(A/C)(T/C) in the promoter regions, and the number of elements are 3 (4CL2, CAD5 and CesA7), 4 (CCOAOMT1 and COMT1) and 5 (CesA8) (Supplementary Figure 1). Then luciferase (LUC) activity assay was used into this experiment. The binary vector which containing a CaMV 35S promoter and the CDS sequence of PtrMYB121was used as the effector, while the vectors containing the six individual promoter fragments and LUC reporter genes were used as reporters (Figure 7A).” “The full-length CDS of PtrMYB121 was amplified by PCR amplification with gene-specific primers (Supplementary Table S1), and ligated into the plant binary vector pCAMBIA1302 driven by the CaMV 35S promoter, and the construct was used as effector for LUC activity assay. The promoter fragments (4CL2 1656bp, CAD5 1733bp, CCoAOMT1 1687bp, COMT 1673 bp, CesA7 1466bp and CesA8 1784 bp) were independently cloned (Supplementary Table S1) and ligated into the pGreen-0800-35mini vector to produce various LUS reporters, and these constructs were used as effectors. All these vectors were individually transformed into the A. tumefaciens strains GV3101.”

So my conclusion is to accept the manuscript after detailed analysis of binding sites or promoter analyses and improving the discussion part. Some other examples of effect of overexpression of SCW component biosynthetic genes should be given.

Other comments:

Did the authors check also trichome density? (Even if there was no significant difference in phenotypes of wt and transgenic plants, some table comparing different parameters could be shown in Supplementary material)

Thank you for your comments. We checked the trichome density, and there was no significant difference in WT and transgenic plants. We have added some information about phenotypes of WT and transgenic plants in Supplementary Table 2, including trichome density, height, diameter, and leaf area/ per rosette leaf.

50mg/l hygromycin is quite a high dose for Arabidopsis, usually 15-20 is sufficient.

Thank you for your comments. We also used 20mg/ml hygromycin at the beginning of our experiment, and found many survival lines were not transgenic Arabidopsis. Therefore, we increased the concentration of hygromycin to 50mg/ml in the later experiments. Maybe this will cause the death of some genetically modified lines, but Increase the positive rate. We will try to find a balance between the positive rate and the transformation efficiency.

English should be checked, especially verbs.

Thank you for your comments. We have checked the English and modified them according to our ability.

Avoid using abbreviances without explaining at least at once (SCW etc…)

Thank you for your suggestion. We have checked the paper, and reduced the abbreviations.

Figs: error bars seem to be the same or very regular (could be possible, but I have another experiences from my own qPCR data)

  Thank you for your comments. We have checked our raw data, and found the error bars are different in Figures (2, 3 and 6). One reason for the situation may be the algorithmic difference of software. We used the software Graphpad Prism 6 to draw figures and the error bars seemed to be regular than the EXCEL 2019. Another reason is perhaps that we compressed the pictures to meet the requirements of the journal, and the pictures were distorted.

Reviewer 3 Report

The manuscript of Liu and colleagues reports on the biological functions of poplar MYB TFs using a functional analysis in an heterologous system, Arabidopsis thaliana. These MYB are involved in SCW formation and seem to be involved in stem patterning. While most experiments were well conducted, some of them require further details or additionnal data. Moreover, the functions of the described MYB (PtrMYB121) were already investigated in a previous article. Therefore, this manuscript reports incremental conclusions about PtrMYB121. These conclusions may be beneficial for the community, but the following points need to be adressed before acceptance:

  • Line 24: RT-qPCR does not allow to conclude about the expression level of one gene as compared to another one.
  • L. 27: there is no data supporting thicker SCW in lines overexpressing PtrMYB121
  • L. 50: secondary CesAs are not only involved in fiber SCW, but also vessels and xylem parenchyma. Please correct.
  • L. 107: MYB170/216 rather than MYB10/216.
  • L. 127: it will be more accurate to refer as "expression of PtrMYB121/55 was lower in leaves... than in stem", RT-qPCR does not allow to conclude about "extremely low expression".
  • L. 130: please modify to "PtrMYB121 was used to further research due to its preferential expression in stem".
  • L. 138: delete "35 S", as you mention the overexpression.
  • L. 144: the authors must indicate the gene expression fold change of each overexpressing line as compared to the WT.
  • L. 147: Please explain why you used two WT lines, and if they have the same characteristics (genetic background, growing conditions...)
  • L 156-157: The authors must provide a statistical analysis based on the number of xylem cell layers in several inflorescence stems from several transgenic lines. How many samples were used for this analysis?
  • Figure 4: The authors must describe the phenotypes of several overexpressing lines. It seems that Fig 4 is only based on a single line, and the reader does not know which one.
  • L. 171: Please be more precise and provide at least the name of the method used for each cell wall component and references for the protocols. The reader does not have to rely on other articles to know which methods were used.
  • L. 203: please rephrase PtrMYB121-overexpressing
  • L. 210: The transcriptional activity of PtrMYB121 was already demonstrated with promoters of genes involved in SCW biosynthesis (including cellulose, xylan and lignin) in poplar (Zhong et al., Plant Physiology, November 2011, Vol. 157, pp. 1452–1468). These results are therefore not novel and merely confirm previous experiments.
  • L. 256: Thicker SCW not demonstrated by experimental data.
  • L. 259: Please be more explicit about these differences with MYB170/216.
  • L. 269: please correct "combing"
  • L.322: The authors should explain how they cloned the various promoter regions used in LUC assay in the suitable plasmids.

Author Response

The manuscript of Liu and colleagues reports on the biological functions of poplar MYB TFs using a functional analysis in an heterologous system, Arabidopsis thaliana. These MYB are involved in SCW formation and seem to be involved in stem patterning. While most experiments were well conducted, some of them require further details or additionnal data. Moreover, the functions of the described MYB (PtrMYB121) were already investigated in a previous article. Therefore, this manuscript reports incremental conclusions about PtrMYB121. These conclusions may be beneficial for the community, but the following points need to be adressed before acceptance:

Line 24: RT-qPCR does not allow to conclude about the expression level of one gene as compared to another one.

Yes, RT-qPCR does not allow to conclude about the expression level of one gene as compared to another one. Thank you for your comments. We have changed the description of the sentence “the expression of PtrMYB121 was significantly up-regulated in the process”

L.27: there is no data supporting thicker SCW in lines overexpressing PtrMYB121

Yes, we didn't supply data to support thicker SCW in lines overexpressing PtrMYB121. Therefore, we checked and deleted the description about thicker SCW through the article.

L.50: secondary CesAs are not only involved in fiber SCW, but also vessels and xylem parenchyma. Please correct.

Thank you for your suggestion. We have added the vessels and xylem parenchyma in the sentence “type II, concerning in the fiber, vessels, and xylem parenchyma synthesis of SCW”.

L.107: MYB170/216 rather than MYB10/216.

  Thank you for your suggestion. It is our mistake. We have changed it.

L.127: it will be more accurate to refer as "expression of PtrMYB121/55 was lower in leaves... than in stem", RT-qPCR does not allow to conclude about "extremely low expression".

Thank you for your suggestion. We have changed it. “the expression of PtrMYB55 and PtrMYB121 was lower in young leaves, old leaves, and roots than in stems”

L.130: please modify to "PtrMYB121 was used to further research due to its preferential expression in stem".

Thank you for your suggestion. We have changed it. “PtrMYB121 was used to further research due to its preferential expression in stem”

L.138: delete "35 S", as you mention the overexpression.

Thank you for your suggestion. We have deleted "35 S".

L.144: the authors must indicate the gene expression fold change of each overexpressing line as compared to the WT.

Thank you for your suggestion. We have added the fold changes of each overexpressing line as compared to the WT.

L.147: Please explain why you used two WT lines, and if they have the same characteristics (genetic background, growing conditions...)

Thank you for your comments. In this paper, we used two WT lines which have the same characteristics (genetic background, growing conditions...). Since these lines are planted in different petri dishes, we want to exclude the influence of petri dishes on the growth of Arabidopsis seedlings. Therefore, we used two WT lines, and found their growth and expression profiles had no significant change.

L. 156-157: The authors must provide a statistical analysis based on the number of xylem cell layers in several inflorescence stems from several transgenic lines. How many samples were used for this analysis?

Thank you for your comments. We have added a description for the calculation of the number of xylem cell layers. “To quantitatively analyze the changes in xylem of PtrMYB121-overexpressing transgenic Arabidopsis, we made the anatomical cross-sections of inflorescence stems and took pictures with an optical microscope system. Then the Image J software was used for calculating the width of xylem area, and every samples had 30 slices for analysis.”

Figure 4: The authors must describe the phenotypes of several overexpressing lines. It seems that Fig 4 is only based on a single line, and the reader does not know which one.

Thank you for your comments. the PtrMYB121-overexpressing lines (L1, L4 and L6) had the similar phenotypes, and in Fig 4 we used the L1. We had added the description “The three independent lines (L1, L4 and L6) had similar phenotypes, and the line L1 which had the highest expression was used for Histochemical staining.”

L.171: Please be more precise and provide at least the name of the method used for each cell wall component and references for the protocols. The reader does not have to rely on other articles to know which methods were used.

Thank you for your suggestion. We have added the description for methods of cell wall component “The acetyl bromide (AcBr) method was used to estimate the total lignin content, while the Van Soest method was used to measure the cellulose and hemicellulose content.”

L.203: please rephrase PtrMYB121-overexpressing

Thank you for your suggestion. We have rephased the sentence “The binary vector which containing a CaMV 35S promoter and the CDS sequence of PtrMYB121 was used as the effector, while the vectors containing the six individual promoter fragments and LUC reporter genes were used as reporters”

L.210: The transcriptional activity of PtrMYB121 was already demonstrated with promoters of genes involved in SCW biosynthesis (including cellulose, xylan and lignin) in poplar (Zhong et al., Plant Physiology, November 2011, Vol. 157, pp. 1452–1468). These results are therefore not novel and merely confirm previous experiments.

Yes, the transcriptional activity of PtrMYB121 was already demonstrated with promoters of genes in previous studies (Zhong et al., 2011). These results are indeed not novel. In this study, we wanted to use the similar results to demonstrate that the phenotypes (width xylem, increased lignin and cellulose) caused by PtrMYB121 due to its activation to the downstream genes (cellulose and lignin). Therefore, the results were not novel, but it could explain the changes caused by PtrMYB121.

L.256: Thicker SCW not demonstrated by experimental data.

Yes, we didn't supply data to support thicker SCW in lines overexpressing PtrMYB121. Therefore, we checked and deleted the description about thicker SCW through the article.

L.259: Please be more explicit about these differences with MYB170/216.

Thank you for your comments. We have added the description about these differences with MYB170/216 “PtoMYB170 positively regulates the lignin deposition and confers drought tolerance, while PtoMYB216 only promoted the accumulation of lignin”

L.269: please correct "combing"

Thank you for your suggestion. We have corrected "combing" with “combining”.

L.322: The authors should explain how they cloned the various promoter regions used in LUC assay in the suitable plasmids.

Thank you for your comments. We have added the detail information of the luciferase assay in “Materials and Methods” and “Results”. “we made the Cis-element analysis of the promoters of lignin (4CL2, CCOAOMT1, COMT1 and CAD5) and cellulose (CesA7 and CesA8) biosynthetic genes. They all had the secondary wall MYB-responsive elements ACC(A/T)A(A/C)(T/C) in the promoter regions, and the number of elements are 3 (4CL2, CAD5 and CesA7), 4 (CCOAOMT1 and COMT1) and 5 (CesA8) (Supplementary Figure 1). Then luciferase (LUC) activity assay was used into this experiment. The binary vector which containing a CaMV 35S promoter and the CDS sequence of PtrMYB121was used as the effector, while the vectors containing the six individual promoter fragments and LUC reporter genes were used as reporters (Figure 7A).” “The full-length CDS of PtrMYB121 was amplified by PCR amplification with gene-specific primers (Supplementary Table S1), and ligated into the plant binary vector pCAMBIA1302 driven by the CaMV 35S promoter, and the construct was used as effector for LUC activity assay. The promoter fragments (4CL2 1656bp, CAD5 1733bp, CCoAOMT1 1687bp, COMT 1673 bp, CesA7 1466bp and CesA8 1784 bp) were independently cloned (Supplementary Table S1) and ligated into the pGreen-0800-35mini vector to produce various LUS reporters, and these constructs were used as effectors. All these vectors were individually transformed into the A. tumefaciens strains GV3101.”

Round 2

Reviewer 2 Report

I am very glad that the authors improved the manuscript by following all the comments. Nevertheless, I still have few minor comments:

row 107: „conserved with AtMYB61 and PtoMYB170/216, and only had the differences with a few bases“ – Please, be concrete.

row 219: „They all had the secondary wall MYB-responsive elements ACC(A/T)A(A/C)(T/C)“ – Please, add some citation.

row 223: „which containing“ - contain

In the discussion part, there could be more papers describing overexpression of other poplar transcription factors or  lignin or SCW-related genes in another species. For example, Ye et al: Over-expression of transcription factor ARK1 gene leads to down-regulation of lignin synthesis related genes in hybrid poplar ‘717’ published in Scientific Reports volume 10, 8549 (2020) is missing or paper of doi: 10.1371/journal.pone.0174748 or DOI: 10.1111/tpj.13112 could be mentioned (just example, I have no personal interests in these papers).

Author Response

I am very glad that the authors improved the manuscript by following all the comments. Nevertheless, I still have few minor comments:

row 107: „conserved with AtMYB61 and PtoMYB170/216, and only had the differences with a few bases“ – Please, be concrete.

Thank you for your suggestion. We have added the description about “only had the differences with a few bases” “The amino acid residues in the R2- MYB domain were changed from histidine to tyrosine, from asparagine to arginine and serine, while the amino acid residues in the R3- MYB domain were changed from alanine to threonine, from isoleucine to leucine (Figure 1A).”

row 219: „They all had the secondary wall MYB-responsive elements ACC(A/T)A(A/C)(T/C)“ – Please, add some citation.

Thank you for comments. We have added the reference for “the secondary wall MYB-responsive elements ACC(A/T)A(A/C)(T/C)”

row 223: „which containing“ - contain

Thank you for comments. We have corrected it.

In the discussion part, there could be more papers describing overexpression of other poplar transcription factors or lignin or SCW-related genes in another species. For example, Ye et al: Over-expression of transcription factor ARK1 gene leads to down-regulation of lignin synthesis related genes in hybrid poplar ‘717’ published in Scientific Reports volume 10, 8549 (2020) is missing or paper of doi: 10.1371/journal.pone.0174748 or DOI: 10.1111/tpj.13112 could be mentioned (just example, I have no personal interests in these papers)

Thank you for comments. We have added some information and references in the discussion part with some recent studies “In addition to Arabidopsis, the R2R3-MYB TFs also involved in the regulation of SCW biosynthesis in other herbs. For example, overexpression of SbMyb60 not only impacts phenylpropanoid biosynthesis, but also alters SCW composition in Sorghum bicolor. In rice, OsMYB103L plays an important role in GA-mediated SCW synthesis, and functions as a master switch of downstream TFs. BdSWAM1 is a positive regulator of SCW thickening in Brachypodium distachyon, and interacts with cellulose and lignin gene promoters.” “Recently, some other TFs also participate in the process of SCW biosynthesis in addition to R2R3-MYB TFs. The overexpression of a pine Dof TF PpDof5 in hybrid poplars promoted the accumulation of lignin in stems and basal leaves. The ARK1 gene from ARBORKNOX family reduced the lignin content by down-regulating lignin synthesis related genes. KNAT 2/6b, one member of the class I KNOX genes, inhibited the xylem differentiation by regulating NAC TF in poplar. The overexpression of PdOLP1, an Osmotin-Like protein gene in poplar, caused reduction of the radial width and cell layer number in xylem and phloem zones, and down-regulated expression of genes involved in lignin biosynthesis, and in the fibers and vessels of xylem cell walls.”

Reviewer 3 Report

The authors have adressed most of the comments, but they did not correctly answered to one of my questions: in paragraphe 2.4 (line 157, stem phenotype of overexpressing line), the reader still does not know how many independant samples were used for the analysis. 30 cross sections of a single stem are not biological replicates. In addition, a statistic analysis between the OE line and the WT is still missing, especially because a single OE line is being used. I'll therefore suggest that the authors adequately adress this point before publication.

Author Response

The authors have adressed most of the comments, but they did not correctly answered to one of my questions: in paragraphe 2.4 (line 157, stem phenotype of overexpressing line), the reader still does not know how many independant samples were used for the analysis. 30 cross sections of a single stem are not biological replicates. In addition, a statistic analysis between the OE line and the WT is still missing, especially because a single OE line is being used. I'll therefore suggest that the authors adequately adress this point before publication.

Thank you for your comments. In the last revision, we did not fully understand what you mean, and couldn't correctly answer this question. In this experiment, we both used 3 independent samples for 40-day-old WT (WT-1) and PtrMYB121-overexpressing Arabidopsis (L1) seedlings. We performed a statistical analysis with one-way ANOVA followed by Dunnett’s test. The description and results are listed as follows:

“The results showed that overexpression of PtrMYB121 caused the significant increasing of cell layers (7-16 layers, Figure 4E and F, Table 1) in xylem compared with WT stems (4-9 layers, Figure 4A and B, Table 1). To quantitatively analyze the changes in xylem of PtrMYB121-overexpressing transgenic Arabidopsis, we further made the anatomical cross-sections of inflorescence stems and took pictures with an optical microscope system. Then the Image J software was used for calculating the cell layer number and radial width of xylem. Both WT (WT-1) and PtrMYB121-overexpressing Arabidopsis (L1) contained 3 independent samples, and every samples had 30 slices for analysis. The cell layers of xylem increased 89.73%-120.84% in PtrMYB121-overexpressing Arabidopsis (Figure 4E and F, Table 1). Xylem area of PtrMYB121-overexpressing transgenic Arabidopsis (Figure 4E and F, Table 1) was 38.35%-64.12% winder than that of inflorescence WT stems (Figure 4A and B, Table 1). Statistical analysis with one-way ANOVA showed the cell layer and radial width of xylem in inflorescence stems had significant differences between WT and PtrMYB121-overexpressing transgenic plants (Table 1), but the 3 independent samples among WT or PtrMYB121-overexpressing transgenic Arabidopsis had no significant difference (Table 1).”

Table1. Cell layer number and radial width of xylem in inflorescence stems of wild type (WT-1) and PtrMYB121-overexpressing transgenic (L1) Arabidopsis plants.

Line

Samples

Cell layer (n)

Radial width (μm)

WT-1

1

6.78 ± 0.81a

439.72 ± 28.79a

2

7.11 ± 0.94a

461.77 ± 33.47a

3

6.43 ± 0.87a

422.36 ± 30.16a

1

13.76 ± 1.57b

657.42 ± 53.34b

PtrMYB121-L1

2

14.20 ± 1.72b

693.17 ± 58.38b

3

13.49 ± 1.55b

638.84 ± 49.67b

Different letters above bars represent statistically significant differences between groups (P < 0.05) as determined by one-way ANOVA followed by Dunnett’s test.
